

# Preferential Pathways for Fluid and Solutes in Heterogeneous Groundwater Systems: Self-Organization, Entropy, Work

1) Erwin Zehe, 1) Ralf Loritz, 2) Yaniv Edery, 3) Brian Berkowitz

1) Karlsruhe Institute of Technology (KIT), Institute of Water and River Basin Management, Karlsruhe, Germany; 2) Technion Israel Insstitute of Technology, Haifa, Israel; 3) Department of Earth and Planetary Sciences, Weizmann Institute of Science, Rehovot, Israel

Corresponding author: Erwin Zehe (Erwin.Zehe@kit.edu)

## Abstract

Patterns of distinct preferential pathways for fluid flow and solute transport are ubiquitous in heterogeneous, saturated and partially saturated porous media. Yet, the underlying reasons for their emergence, and their characterization and quantification, remain enigmatic. Here we analyze simulations of steady state fluid flow and solute transport in two-dimensional, heterogeneous saturated porous media with a relatively short correlation length. We demonstrate that the downstream concentration of solutes in preferential pathways implies a downstream declining entropy in the transverse distribution of solute transport pathways. This reflects the associated formation and downstream steepening of a concentration gradient transversal to the main flow direction. With an increasing variance of the hydraulic conductivity field, stronger transversal concentration gradients emerge, which is reflected in an even smaller entropy of the transversal distribution of transport pathways. By defining "self-organization" through a reduction in entropy (compared to its maximum), our findings suggest that a higher variance and thus randomness of the hydraulic conductivity coincides with stronger macroscale self-organization of transport pathways. While this finding appears at first sight striking, it can be explained by recognizing that emergence of spatial self-organization requires, in light of the second law of thermodynamics, that work be performed to establish transversal concentration gradients. The emergence of steeper concentration gradients requires that even more work be performed, with an even higher energy input into an open system. Consistently, we find that the energy input necessary to sustain steady-state fluid flow and tracer transport grows with the variance of the hydraulic conductivity field as well. Solute particles prefer to move through pathways of very high power, and these pathways pass through bottlenecks of low hydraulic conductivity. This is because power depends on the squared spatial head gradient, which is in these simulations largest in regions of low hydraulic conductivity.





## 1 Introduction

### 1.1 Preferential flow phenomena – fast, furious and enigmatic

Distinct patterns of preferential movement of water, dissolved and suspended matter are ubiquitous in fully-saturated aquifer systems (e.g., LaBolle and Fogg, 2001; Bianchi et al., 2011; Berkowitz et al., 2006), partially saturated soils (e.g., Beven and Germann, 1982) and at the land surface (e.g., Uhlenbrook, 2006). Preferential flow and solute transport in porous media commonly leads to fast, localized early arrivals and/or long tailing in temporal breakthrough curves (e.g., Berkowitz et al., 2006) and pronounced fingerprints in concentration patterns in soils (Flury et al., 1994).

Preferential flow and transport often occur along connected highly conductive networks. Some networks are formed by previous physical/chemical work performed by the fluid, as in the cases of surface rill and river networks (Howard, 1990), subsurface pipe networks (Jackisch et al., 2017), karst conduits (Groves and Howard, 1994), and fractured rock formations (Becker and Shapiro, 2000; Berkowitz, 2002). Other networks are created by soil fauna and flora as earth worm burrows (Zehe and Flühler, 2001; van Schaik et al., 2014) and plant roots (Wienhöfer et al., 2009; Tietjen et al., 2009). Although it might appear unsurprising that flow and transport through these networks dominates system behavior, effective ways to model flow and transport in these networks have been debated for more than 30 years (Beven and Germann, 1981; Šimůnek et al., 2003; Klaus and Zehe, 2011; Wienhöfer and Zehe, 2014; Berkowitz et al., 2006, Sternagel et al., 2019, 2020). Preferential flow and transport occurs, however, also in porous media without such "well-defined" networks, e.g., in coarse-grained soils due to fingering and wetting front instabilities (Blume et al., 2009; Dekker and Ritsema, 2000; Ritsema et al., 1998) and particularly in stochastically heterogeneous saturated porous media (Bianchi et al., 2011; Edery et al. 2014).

The emergence of preferential pathways in systems without well-defined networks – and their characterization – remains even more enigmatic. The numerical study of Edery et al. (2014), for example, revealed that a higher variance in the hydraulic conductivity ($K$) field coincided with a stronger concentration of solutes within a smaller number of preferential flow paths. If the emergence of preferential flow is indeed manifested self-organization, as argued by Berkowitz and Zehe (2020), this key finding of Edery et al. (2014) suggests that macroscale steady states of stronger organization (or higher order) emerge and persist despite a greater





degree of subscale randomness. The related key questions we address here are (i) how spatial
organization in preferential fluid flow and solute transport can be quantified, and (ii) why a
larger subscale randomness might favor stronger macroscale organization.

### 1.2 Attempts to characterize and predict preferential transport in groundwater

The emergence of preferential pathways of fluid flow and solute transport in saturated porous
media has been explored in numerous simulation studies in heterogeneous conductivity fields,
to relate the spatial correlation structures of the hydraulic conductivity and velocity fields to
features of anomalous transport behavior (e.g., Cirpka and Kitanidis, 2000; Willmann et al.,
2008; Berkowitz and Scher, 2010; de Dreuzy et al., 2012; Morvillo et al., 2021). While velocity
correlation parameters have been successfully related to statistical moments of hydraulic
conductivity, it remains challenging or even impossible to a priori delineate preferential
pathways exclusively based on multivariate and topological characteristics of the hydraulic
conductivity field. Cirpka and Kitanidis (2000) and Willmann et al. (2008) report, for instance,
the emergence of preferential pathways in the distributions of tracer travel velocities and shapes
of solute plumes. These pathways were not apparent, however, from the analysis of the
stationary conductivity fields. Moreover, Edery et al. (2014) demonstrate that critical path
analysis (based on percolation theory), for example, does not determine the actual preferential
pathways in a system; the authors suggest that the operational preferential pathways become
fully apparent only when solving for fluid flow and solute transport through the domain.
Bianchi et al. (2011) explored the link between connectivity and the emergence of preferential
flow paths at the MADE site, using three-dimensional, conditional, geostatistical realizations
of the hydraulic conductivity field. Their simulations of flow and transport under permeameter-
like boundary revealed that the first 5% of particles, arriving at the downstream domain outlet,
moved through preferential flow paths carrying 40% of the flow. Fiori and Jankovic (2012)
reported similar findings and stressed the rather small probability that solute particles visit
highly conductive blocks particularly in case of a high variance in $K$. Bianchi et al. (2011)
highlighted that the fraction of particle paths passing the high-conductivity regions was between
43% and 69%, while the most rapid transport passed through low-conductivity bottle necks.
This is in line with the findings of Edery et al. (2014), who concluded that connectivity of rapid
preferential pathways need not require connected zones of continuously high hydraulic
conductivity. Along a different avenue, Bianchi and Pedretti (2017) characterized spatial



disorder in two-dimensional conductivity fields by means of the Shannon entropy (Shannon,
1948) and related this to moments of solute breakthrough curves. Dispersion in travel times and
the probability of solutes to pass through low conductivity regions were found to increase with
lower order expressed by a higher geological entropy.
**1.3 Preferential flow, self-organization, entropy, work – where is the connection?**
The results of the studies mentioned above all underpin that (a) preferential flow and transport
in heterogeneous, saturated porous media remains a largely enigmatic and emergent
phenomenon, and (b) efforts to represent this behavior by means of effective transfer functions,
inferred from volume-averaging based scaling of the hydraulic conductivity field, appear
virtually impossible. This is why, we propose to shift the attention from the question of "where"
preferential pathways emerge, to questions regarding their "macroscale organization and
strength", and "the necessary physical work" to establish their self-organized emergence.
Haken (1983) defined self-organization as the emergence of ordered macroscale states, or the
dynamic behavior of an open system far from thermodynamic equilibrium (TE), that arises from
a synergetic interplay of microscale, irreversible processes. An ordered state is characterized
by the deviation of its entropy from the entropy maximum at TE (Kondepudi and Prigogine,
1998, see section 3). This reduction in entropy, and any additional entropy produced by the
internal irreversible processes, must be exported from the open system to establish order. This
is turn requires physical work, and thus an input of free energy into the system, that is large
enough to create and maintain the self-organized state. A classical example to illustrate that
self-organization in open systems requires free energy and work, which inspired also Haken's
theory of "synergetics", is a gas laser. Laser light results from coherent stimulated light
emissions from all molecules in the system. Stimulated emission emerges when the energy input
into the gas laser becomes sufficiently large that the number of stimulated molecules exceeds
the number of molecules in the basic state. This "energetic pumping" establishes a state very
far from thermodynamic equilibrium, corresponding to an even apparently negative absolute
temperature in Boltzmann statistics, at which coherent emission from all individual emissions
emerges. Haken (1983) postulated that a higher-order, non-local "enslavement principle" forces
the individual molecules into a coherent and thus ordered behavior. This example of a critical
pumping rate to establish organization of laser light will be shown below (section 4) to be
analogous to fluid flow through porous media.





Several researchers have suggested that self-organization and the formation of complex
organisms and patterns in biological and environmental systems are governed by non-
local/global energetic extremal principles, in analogy to the Haken (1983) enslavement
principle. Pioneering studies in this context proposed that species maximize their energy
throughput (i.e., power) during evolution (Lotka, 1922 a &b) or showed that steady-state
planetary heat transport may be modeled successfully with a very simple two-box model, when
assuming that this state maximizes entropy production (Paltridge, 1979). This work motivated
several studies that explored the possibility that energetically optimized model setups allow
hydrological prediction of the land surface energy balance and evaporation (Kleidon et al.,
2014; Zehe et al., 2013), rainfall runoff behavior (Zehe et al., 2013) and groundwater flow and
spring discharge (Hergarten et al., 2014). These and other studies generally showed that
preferential flow in connected networks allows for a more energy efficient throughput of water
and matter through the system. This is because they reduce flow-weighted dissipative losses
due to an increased hydraulic radius in the rill or river network compared to sheet overland flow
(Howard, 1990; Kleidon et al., 2013) or in subsurface connected preferential pathways
compared to matrix flow (Hergarten et al., 2014; Zehe et al., 2010).
While the second law of thermodynamic refers to physical entropy (introduced by Clausius
(1857), section 3.1), information entropy (introduced by Shannon (1948)) is closely related and
well suited for diagnosing spatial organization (section 3.3). The concepts of information and
Shannon entropy haven been used widely to characterize irreversible mixing and reaction
processes and their predictability (Chiogna and Rolle, 2017), the emergence of order in
distributed time series (Malicke et al., 2020), information in multiscale permeability data
(Dell`Oca et al., 2020) and the role of spatial variability of rainfall and topography in distributed
hydrological modelling (Loritz et al., 2018, 2021). Woodbury and Ulrych (1993) and Kitanidis
(1994) used the Shannon entropy to describe the spatial-time development and dilution of tracer
plumes in groundwater systems. Chiogna and Rolle (2017) expanded the dilution index for the
case of reactive solute mixing in groundwater and found a critical value that indicated the
complete consumption of a reactant, which was independent of advection and dispersion.
Bianchi and Pedretti (2017) used the Shannon entropy to quantify spatial disorder in
stochastically generated alluvial aquifers and explored its relation to the first three moments of
simulated tracer break through curves. They found the average breakthrough time and its





variance to increase with increasing geological entropy, while the skewness in travel times
declined with increasing geological entropy increasing disorder.

### 1.4 Objectives

We thus suggest that the concepts of entropy, free energy and work hold the key to better
understand why preferential flow and transport in unstructured heterogeneous, saturated porous
media actually emerge. To this end, we analyze simulations of fluid flow and solute transport
through stochastically heterogeneous aquifers with different degrees of randomness (variance
in hydraulic conductivity), based on the results and insights of Edery et al. (2014). Specifically,
we show that macroscale self-organization due to the emergence of preferential solute transport
can be quantified based on the downstream decline of the Shannon entropy of the transversal
concentration pattern. We also find that preferential patterns of higher order, expressed through
lower entropies, emerge in case of larger variances of hydraulic conductivity. What appears
almost as a paradox at first sight – in the sense that a higher subscale randomness of the medium
causes a stronger spatial organization – can be explained by the fact that the power required to
maintain the driving head difference in steady state increases with increasing variance of the
hydraulic conductivity field. Due to this higher energy input, the fluid and solutes may perform
the necessary work to form preferential transport pathways that pass rapidly through low
conductivity bottlenecks and form preferential flow paths by steepening transversal
concentration gradients. We show, finally, that the entropy in the corresponding breakthrough
curve (BTC) increases with the variance of the hydraulic conductivity. This can be explained
by recognizing that entropy cannot be consumed, due to the second law of thermodynamics.
Hence, the downstream declining entropy in the transversal distribution of solute needs to be
exported from the system, and this export is reflected in the higher entropy of the corresponding
BTC.

### 2 Underlying simulations of fluid flow and transport

### 2.1 Media generation and numerical simulations of fluid flow

Here, we partially revisit and expand upon the numerical simulations of Edery et al. (2014),
which were employed to provide insight on fluid flow and anomalous solute transport behavior.
Edery et al. (2014) considered steady-state fluid flow within a two-dimensional, stochastic
heterogeneous system. The flow domain measured 300 by 120 space units as discretized into





grid cells of uniform size $\Delta x = 0.2$, $\Delta y = 0.2$, while all quantities are expressed using the same
space-time units. We consider a deterministic head difference of 100, from the left (vertical)
upstream boundary to the right downstream boundary; no-flow conditions are assigned to the
two horizontal domain boundaries.
We generated random realizations of statistically homogeneous, isotropic Gaussian fields for
the natural logarithm of the hydraulic conductivity $\ln(K)$, with exponential covariance and mean
$\ln(K) = 0$, using the sequential Gaussian simulator GCOSIM3D (Gómez-Hernánez et al,. 1997).
Edery et al. (2014) considered fields associated with a unit correlation length, $l = 1$, exploring
the impact of different values of the variance of $\ln(K)$, i.e., $1 < \sigma^2 < 5$, on the emergence of
preferential solute transport.
Figure 1a shows a realization for $\sigma^2 = 3$, corresponding to mild and strong randomness for
distances larger than $3l$. The deterministic flow problem for each realization was solved using
a code that is based on finite elements with Galerkin weighting functions (Guadagnini and
Neuman, 1999). The corresponding hydraulic head values throughout the domain were
converted to local velocities, and thus streamlines (Fig. 1b), which were in turn used for
transport simulations using particle tracking. For the system considered here, we used a porosity
of 0.3 (e.g., Levy and Berkowitz, 2003).

**2.2 Simulated solute transport with particle tracking**

Solute movement in each domain realization was simulated using the calculated streamlines,
with a standard Lagrangian particle tracking method. For all domains, values of $\Delta$ and $l$ were
chosen such that $l/\Delta = 5$, to enable capture of small-scale fluctuations and advective transport
features (Ababou et al., 1989; Riva et al., 2009). Along the left upstream boundary, particles
are injected, by flux-weighting, and advance by advection and diffusion. The Langevin equation
defines the particle displacement vector **r**, starting from given particle locations at time $t_k$:
$$\mathbf{r} = \boldsymbol{v}[\mathbf{x}(t_k)]\delta t + \boldsymbol{d}_o \quad (\text{Eq. 1})$$
where $\boldsymbol{v}$ is the fluid velocity vector, $\delta t$ is the time step magnitude, and $\boldsymbol{d}_o$ denotes the diffusive
displacement, with a modulus of $\boldsymbol{d}_o$ given by $\xi\sqrt{2D_{\mathrm{mol}}\delta t}$; $\xi$ is a random number drawn the from



standard normal distribution $N[0, 1]$.   A representative molecular diffusion coefficient of
$D_{mol} = 10^{-9}$  m$^2$   s$^{-1}$  was prescribed (Domenico and Schwartz, 1990). The advective
displacements in Equation 1 are computed using the local velocities at **x** with a fixed, uniform
spatial step $\delta s$. In Equation 1, the time step $\delta t$ is given by $\delta t = \delta s / v$, where $v$ is the modulus of
**v**. Reflection conditions are prescribed along the two horizontal no-flow boundaries to avoid
particle leakage. As in Edery et al. (2014), we used $10^5$ particles, with $\delta s = \Delta / 10$.

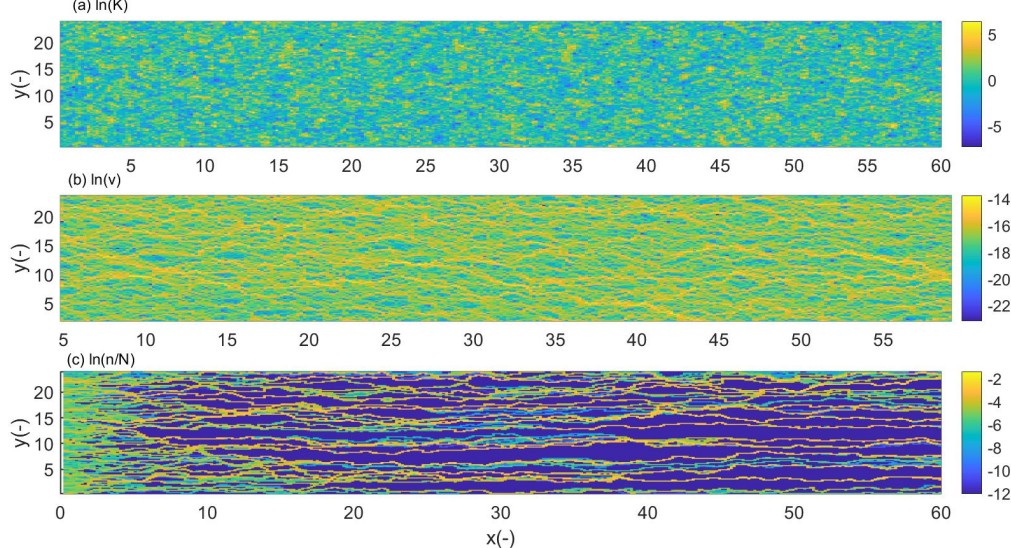


Figure 1: Examples of (a) $\ln(K)$, (b) $\ln(v)$, and (c) the cumulative number of particles that visited
a grid cell in the simulation domain, normalized with the total number of particles N, on a
logarithmic scale. The variance of $\ln(K)$ is $\sigma^2 = 3$.
**3 Free energy, entropy and work**
**3.1 Thermodynamics in a nutshell: the first and the second law**
We start very generally with the first law of thermodynamics, which relates the variation in
internal energy $U$ (J = kg m$^2$ s$^{-2}$) of a system to a variation of work $E_{free}$ (J) and a variation of
heat $Q_h$ (J), while overall energy is conserved:
$$\delta U = \delta E_{free} + \delta Q_h \ (Eq.\,2)$$





Note that the capacity of a system to perform work is equivalent to "free energy", while a
variation in heat is equal to the product of a variation of physical entropy $S$ (J K$^{-1}$) and the
absolute temperature $T$ (K): $\delta Q_h = T\,\delta S$ as introduced by Clausius (1857). The second law of
thermodynamics states that entropy is produced during irreversible processes, while it cannot
be consumed. The second law implies that isolated systems, which neither exchange mass, nor
energy, nor entropy with their environment, reach a dead state of maximum entropy called
thermodynamic equilibrium in which all gradients have been depleted. Kleidon (2016)
distinguishes three types of physical entropy: thermal entropy produced by friction and
depletion of temperature gradients, molar entropy produced by mixing and depletion of
chemical potential/concentration gradients, and radiation entropy produced by radiative cooling
and depletion of radiation temperature differences.
From Eq. 2 and the second law, we can conclude that free energy is not a conserved property,
as it corresponds to the variation in internal energy minus the variation in heat, during which
entropy is produced. Free energy dissipation and entropy production are thus inseparable, and
maximization of the entropy of an isolated system occurs due to conservation of energy at the
expense of minimizing its free energy. An open system may nevertheless persist in steady states
of lower entropy, if it is exposed to a sufficient influx of free energy to sustain the necessary
physical work that needs to be performed to act against the natural depletion of the internal
gradients, or even to steepen them and further reduce the entropy (as discussed for the gas laser).
Order in an open system thus manifests through persistent gradients and an entropy lower than
the maximum. Steps to higher order and lower entropies imply a steepening of internal
gradients. This is exactly what occurs when preferential transport of solutes emerges in our
transport simulations: solute particles tend to concentrate in localized pathways, thereby
forming a transversal concentration gradient (according to the domain geometry shown in Fig.
1). The Shannon entropy (Shannon, 1948) is ideally suited to quantify the related entropy
reduction, as detailed in section 3.3.

**3.2 The free energy balance of saturated porous media**

To determine the work that is performed by the fluid when flowing through heterogeneous
media, we derive the free energy balance of the fluid by relating changes in hydraulic head and
fluid flux to their energetic counterparts. The local formulation of the free energy balance of a
groundwater system, seen as an open thermodynamic system, is determined by the



difference/divergence of the free energy fluxes $\boldsymbol{J}^E_{free}$ (J s$^{-1}$ m$^{-2}$) per unit area and the amount of
dissipated energy per volume $D$ (J s$^{-1}$ m$^{-3}$):
$$\frac{de_{free}}{dt} = -\nabla \cdot \boldsymbol{J}^E_{free} - D \ (Eq.\,3)$$

where $e_{free}$ (J s$^{-1}$ m$^{-3}$) is the volumetric free energy density. Advective fluxes of relevant free
energy forms are generally determined by multiplying the Darcy flux with the corresponding
form of energy per unit volume. Here we account for advection of mechanical energy $\boldsymbol{J}^E_H$
(named power hereafter), gravitational potential energy $\boldsymbol{J}^E_{pot}$, and kinetic energy of the flowing
fluid $\boldsymbol{J}^E_{kin}$. As energy is additive, the term $\boldsymbol{J}^E_{free}$ corresponds hence to the sum of the following
free energy fluxes:
$$\boldsymbol{J}^E_H = \boldsymbol{q}\rho g H$$

$$\boldsymbol{J}^E_{pot} = \boldsymbol{q}\rho g z \ (Eq.\,4)$$

$$\boldsymbol{J}^E_{kin} = \boldsymbol{q}\frac{1}{2}\rho v^2$$

where $\rho$ (kg m$^{-3}$) is the density of water, $g$ (m s$^{-2}$) the gravitational acceleration, $\boldsymbol{q}$ (m s$^{-1}$) the
Darcy flux, $v$ (m s$^{-1}$) the absolute value of the fluid velocity, $H$ (m) the pressure head, and $z$ (m)
the geodetic elevation. Note that while kinetic energy is proportional to $v^2$, the kinetic energy
flux corresponds to the product of the volumetric water flux $\boldsymbol{q}$ and its kinetic energy density
$\frac{1}{2}\rho v^2$. Thus, kinetic energy is in fact proportional to $v^3$ and is usually very small. By inserting
Eq. 4 into Eq. 3, we obtain:
$$\frac{de_{free}}{dt} = -\rho g \nabla [\boldsymbol{q}(H + z)] - \frac{1}{2}\rho \nabla [\boldsymbol{q}v^2] - D \ (Eq.\,5)$$

The left hand side of Eq. 5 corresponds to the change in Gibbs free energy of a fluid volume
under isothermal conditions (Bolt and Frissel, 1960). This change in free energy storage on the
left hand side can be decomposed into the sum of three terms as well (Zehe et al., 2019): (i) the
change in storage of gravitational potential energy reflecting soil water storage changes in
partially saturated soils or density changes in groundwater; (ii) the change in storage of
mechanical energy reflecting changes in pressure head in groundwater or changing matric
potentials in partially saturated soils; and (iii) the change in kinetic energy stored in the system,
due to acceleration of the fluid. The latter is usually very small and can be neglected.


In the case of steady-state groundwater flow, the variables $H$, $z$, $\rho$ and $v$ are constant in time, so
that the change in free energy storage at the left hand side of Eq. 5 is zero. As we assume $z$ to
be constant along the system and neglect density changes of the fluid, the divergence in the flux
of gravitational potential energy at the right hand side is zero, as well. The system under
investigation hence receives solely steady-state inflow of high mechanical energy,
corresponding to the upstream inflow of water at a high pressure head, and it exports water at
a much lower mechanical energy at the lower downstream pressure head. The corresponding
energy difference is partly dissipated and partly converted into kinetic energy of flowing fluid
and dissolved solute masses. The latter is, however, usually neglected, as dissolved solute mass
is much smaller. As steady-state fluid flow further implies that the divergence of $\boldsymbol{q}$ is zero as
well, the free energy (Eq. 4) becomes hence:
$$\rho g \boldsymbol{q} \cdot \nabla H = -\rho v \boldsymbol{q} \cdot \nabla v - D \ (Eq.\,6).$$
The left hand side is the available power per unit volume $P$ (J s$^{-1}$ m$^{-3}$) in the groundwater flux,
which is partly converted into a spatial change in kinetic energy of the fluid and partly
dissipated. In contrast to overland flow systems (Loritz et al., 2019; Schroers et al., 2021), the
change in kinetic energy can be neglected for groundwater as it is proportional to the cube of
the fluid velocity (as noted before Eq. 5). In fact, the use of Darcy's law implies that kinetic
energy can be neglected.
The total available power $P$ in the groundwater flux during steady-state flow is hence nearly
completely dissipated:
$$P = \rho g \boldsymbol{q} \cdot \nabla H = -D \ (Eq.\,7).$$
By inserting Darcy's law into Eq. 7 and recalling that we focus on a two-dimensional domain,
we obtain an equation that relates power and dissipation to the squared head gradient (in sense
of a scalar product):
$$P = -\rho g K \left[ \frac{\partial H}{\partial x}\frac{\partial H}{\partial x} + \frac{\partial H}{\partial y}\frac{\partial H}{\partial y} \right] = -D \ (Eq.\,8).$$
The physical mechanism that causes dissipation relates to the shear and frictional losses the
fluid experiences when passing through the porous medium. As hydraulic conductivity relates
to the ratio of intrinsic permeability $k$ (m$^2$) and viscosity of the fluid $\eta$ (N sm$^{-1}$), the inverse of



*K* is a measure of the flow resistance and related dissipative losses. One would thus expect that
the dissipative losses grow with fluid viscosity (declining *K*, increasing resistance) and
declining permeability (declining *k*). To better underpin this, we simplify Eq. 8 for steady-state
flow through an heterogeneous, one-dimensional system, which means that $\frac{\partial H}{\partial y}=0$:
$$P = \rho g(K(x)\mathrm{d}_x H)\mathrm{d}_x H = D(x) \ (Eq.\,9).$$
where $\mathrm{d}_x$ denotes the gradient with respect to *x*. Steady-state flow in one dimension implies a
constant flux *q* in the *x* direction, which means that the total spatial variation of d*q* is zero. As
*K* is spatially variable, this implies that local spatial variations of conductivity denoted by
$d(K(x))$ must be compensated by opposite spatial variations of the pressure head gradient,
$d(\mathrm{d}_x H)$:
$$dq = 0 \rightarrow$$
$$d(-K(x)\mathrm{d}_x H) = 0 \rightarrow$$
$$-d\big(K(x)\big)\,\mathrm{d}_x H \ = K(x)\,d(\mathrm{d}_x H) \ \ Eq.\,(10)$$
As a consequence, power *P* is not constant (Eq. 7) but instead grows with the magnitude of
local spatial variations of the head gradient $d(\nabla_x H)$:
$$dP = \rho g q\,d(\mathrm{d}_x H) \ (Eq.\,11\,).$$
Due to Eq. 10 (constant Darcy flux), we can express the spatial variation in the head gradient
$d(\mathrm{d}_x H)$ in Eq. 11 as follows:
$$-\mathrm{d}_x H\,d\big(\ln(K(x)\big) = d(\mathrm{d}_x H) \qquad (Eq.\,12).$$
Combining Eq. 12 with Eq. 11, together with the definition of power in Eq. 9, yields:
$$dP = -P(x)\,d\big(\ln(K(x)\big) \rightarrow d(\ln(P(x)) = -d(\ln\big(K(x)\big)\,(Eq.\,13\,).$$
As a consequence, we expect an anti-proportionality between $\ln(P(x))$ and $\ln(K(x))$ for the one-
dimensional case. In conclusion, we propose that the necessary power to push the fluid through
an heterogeneous medium grows also in the two-dimensional case with the variance of the $\ln(K)$
field. Local areas of high power coincide with large positive deviations from the overall average
head gradient, and these in turn peak across regions of low conductivity. This makes sense, as





dissipation peaks in those areas as flow resistance reach a maximum and the required work to
push fluid through these bottlenecks grows as well. This potentially explains the finding of
Edery et al. (2014) that the preferential flow paths also pass through areas of low conductivity.
We discuss this idea further in section 5.
**3.3 Characterizing emergent spatial organization in solute transport using information**
**entropy**
We now address the connection between physical entropy and information entropy, and explain
how we use the latter to quantify ordered states due to the emergence of preferential flow paths
and the associated formation of a concentration gradient transversal to the main flow direction.
The Shannon entropy $S_H$ (bit) is defined as the expected value of information (Shannon, 1948).
Here we defined $S_H$ using the discrete probability distribution to find particles at a distinct
transversal position $y$ at a given $x$ coordinate, as detailed below.
The field of information theory, originally developed within the context of communication
engineering, deals with the quantification of information with respect to a concept called
"surprise" of an event (Applebaum, 1996). For a discrete random variable $Y$ that can take on
several values $y_i$ with associated prior probabilities $p(y_i)$ the surprise or information content of
receiving/observing a specific value $Y = y_i$ is defined as:

$$I = -\log_b\big(p(y)\big) \quad (Eq. 14)$$

where $I$ is the information content, $b$ is the base of the logarithm and $p(y_i)$ the prior probability
that $Y$ can be observed in the state $y$. Due to the use of the logarithm in Eq. 14, information is
an additive quantity, similar to physical entropy, energy, and mass. The expected information
content associated with the probability distribution of the random variable $Y$ is the Shannon
entropy $S_H$:

$$S_H(Y) = -\sum_{y \in Y} p(y_i)\, log_2\, p(y_i) \quad (Eq. 15)$$

The definition of the Shannon entropy is equivalent to Gibb's definition of physical entropy in
statistical mechanics (Ben-Naim, 2008). The latter is obtained when using the natural logarithm
in Eq. 15 and by multiplying the sum with the Boltzmann constant ($k_B$=1.30640 × 10$^{-23}$ J K$^{-1}$).
Physical entropy describes, in terms of statistical mechanics, the number of microstates that



correspond to the same macro-state at a given internal energy. In the state of maximum entropy
where all gradients are depleted, each microstate is equally likely (Kondepudi and Prigogine,
1998). The probability $p$ of a single state is in this case, hence, simply the inverse of the number
of microstates. This implies a maximum uncertainty about the microstates and corresponds to
a minimum order in the system. Jaynes (1957) transferred this fundamental insight into a
method of statistical inference, stating *"when making inferences based on incomplete*
*information, the best estimate for the probabilities is the distribution that is consistent with all*
*information, but maximizes uncertainty"*. We emphasize that a maximum in information
entropy and physical entropy commonly implies a zero gradient either in probability (from the
information perspective) or in an intensive state variable such temperature, concentration or
pressure (from the thermodynamic perspective).
Its straightforward implementation makes Shannon entropy a flexible means (i) for the
optimization of observation networks (Fahle et al., 2015; Nowak et al., 2012), (ii) for the
characterization of mixing and dilution of solute plumes (e.g., Woodbury and Ulrych, 1993;
Kitanidis, 1994), or (iii) to illuminate how spatial disorder in hydraulic conductivity relates to
statistical moments of solute breakthrough curves (Bianchi and Pedretti, 2017). Here we adopt
a straightforward use of the Shannon entropy to characterize simulated solute transport, as
introduced by Loritz et al. (2018) to characterize redundancy in a distributed hydrological
model ensemble. We suggest that the maximum uncertainty corresponds to the case where each
flow path through the domain is equally likely, and the probability distribution to find particles
in a position in the *y*-direction is, hence, uniform. Deviations from this entropy maximum reflect
spatial order due to the concentration of particles in preferred flow paths and the associated
persistence of a transversal concentration gradient. This can be analyzed by computing the
Shannon entropy of the particle density distributions along $y$, $S_H(x)$, at a fixed position $x$ along
the main flow direction, using the particle density matrix. A state of maximum entropy implies
that the same number of particles has visited each of the 120 grid cells at a given $x$ coordinate
i.e. $S_H^{max} = log_2(120) = 6.9$ bits. A state of perfect spatial organization and zero entropy
arises, on the other hand, when all particles move through a single grid cell at a distinct
coordinate $x$.





**4 Results**
In the following, we demonstrate that preferential transport is indeed manifested self-
organization and showcase that a stronger self-organization requires indeed more physical
work. To this end, we calculated the Shannon entropy of transversal flow paths distribution and
relate this to power in fluid flow across the range of the variances in $\ln(K)$ as detailed below.
For this purpose, we set the dimensionless length and time units to meters and seconds,
respectively.
**4.1 Preferential flow paths and flow path entropy as function of the variance in $\ln(K)$**
Figures 2a-c compare the accumulated particle densities that passed through grid cells in the
domain as a function of the variance, $\sigma^2$, for a randomly selected realization. The solute
transport pathways extend in a largely parallel form and share rather similar particle densities
for $\sigma^2=1$. However, the number of pathways clearly declines with increasing variance, and they
exhibit a stronger meandering and a larger visitation of particles in a smaller transversal number
of grids on their downstream course. The Shannon entropy $S_H$ of the flow paths (flow path
entropy hereafter) exhibits, in general, and for all three variance cases, a clear decline with
increasing downstream transport distance (Figs. 2d-f). This reflects the increasing order in the
flow path distribution, corresponding to the emerging and increasing transversal concentration
gradients. A comparison of $S_H$ among the variance cases clearly corroborates the visual
impression that the number preferential flow paths declines with increasing subscale
randomness, while the concentration of solutes therein increases. The analysis of flow path
entropy within the entire set of 100 realizations revealed that this behavior is not an artefact of
single realization. The flow path entropy average across all realization of a variance case
exhibits a steady downstream decline (Fig. 3 a), and the curves are clearly shifted to lower
values with increasing variance of $\ln(K)$. The boxplots in Fig. 3b characterize the distribution
of $S_H(x)$ at the downstream outlet among the realizations. While the spreading and the skewness
of the distribution clearly increases with increasing variance in $\ln(K)$, we also observe that flow
path entropy at the outlet declines clearly and statistically significantly with increasing variance.

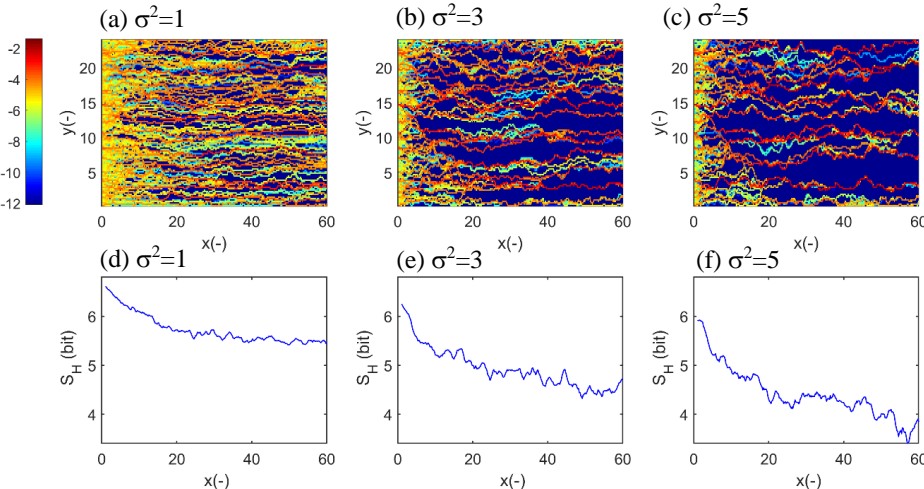

Figure 2: Accumulated, normalized number of particles that passed a distinct point in the
domain as function of the variance in ln($K$), $\sigma^2$, ((a), (b), (c)) and the corresponding Shannon
entropy of the transversal concentration, $S_H$, as a function of the main flow direction ((d), (e),
(f)).

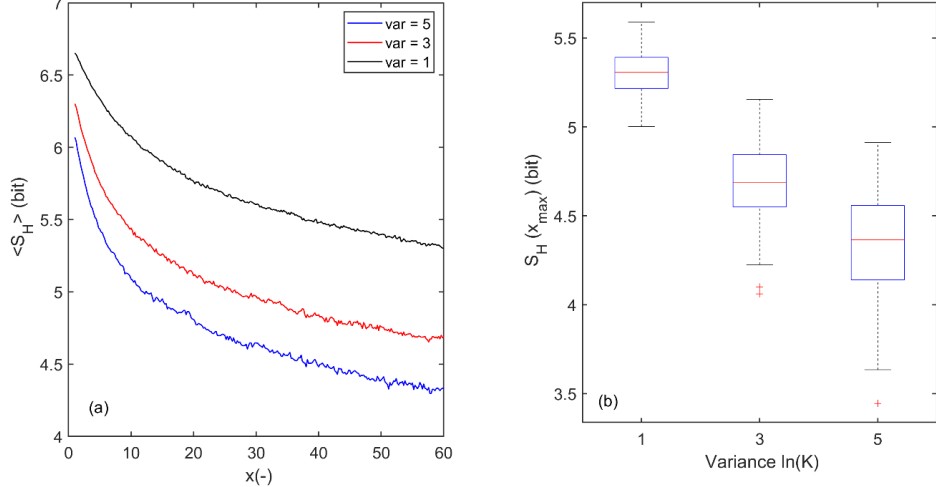


Figure 3: Flow path entropy averaged across all 100 ensemble realizations <$S_H$> as function of
downstream transport distance (a). Boxplot of flow path flow path entropy at the domain outlet
for all realizations of the three variance cases (b); note this corresponds to the asymptotic values
in (a) at x(-) = 60.



We thus state that a higher variance – and thus randomness – in hydraulic conductivity
coincides, for all realizations, with stronger a downstream reduction of the flow path entropy.
This corresponds to a macrostate of higher order due to a more efficient self-organization into
a state of stronger preferential transport.
**4.2 Power in fluid flow as function of the variance in ln($K$)**
Figures 4a-c compare the distribution of power in the fluid flow calculated according to Eq. 7,
as a function of the variance of ln($K$) in the different domains, using a logarithmic scale. For
consistency, we used the same ensemble as for Fig. 2. The distributions of power in the fluid
generally spread across a wide range of magnitudes and are skewed to the left. However, the
distributions clearly shift to larger values and their spread becomes wider when moving to larger
variances.

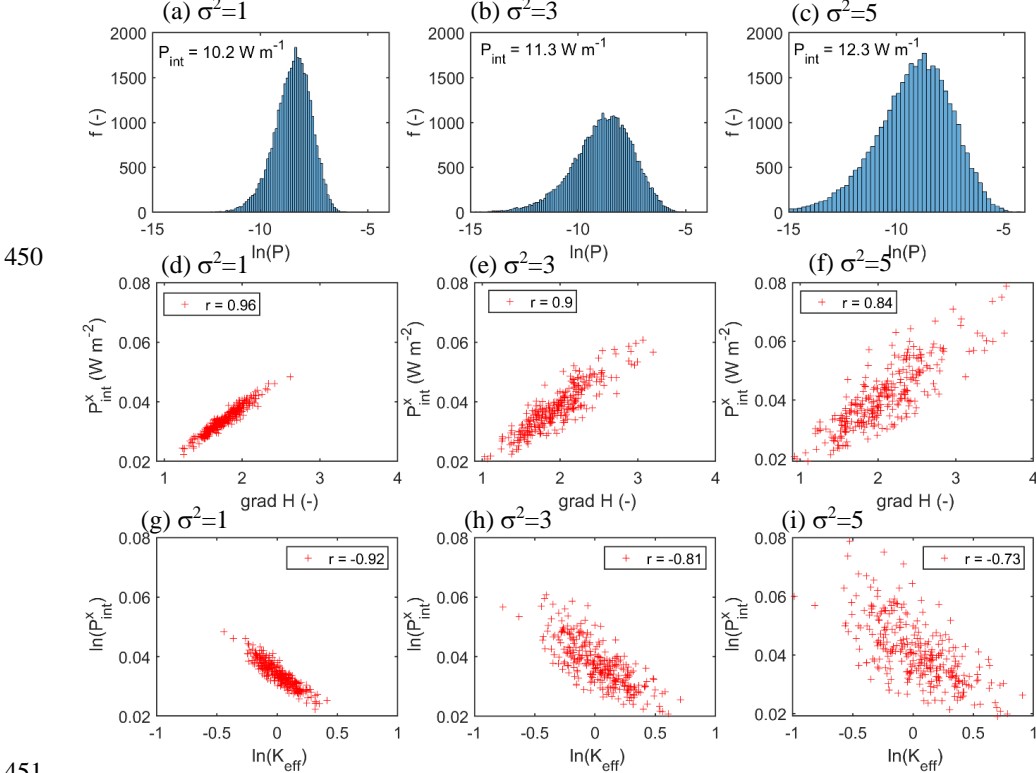

Figure 4: Histogram of ln($P$) as function of the variance $\sigma^2$ ((a), (b) (c)), integral power $P^x_{int}$ in
the total downstream water flux, plotted against the laterally averaged head gradient ((d), (e),
(f)), and ln($P^x_{int}$) as function of the ln of transversally averaged ln($K_{eff}$) ((g),(j), (h)).





This is underpinned when comparing the integrated power in fluid flow across the entire two-
dimensional domain. An increase in variance by two orders of magnitude in the log-normal
scale corresponds to an increase in power of 2 W per unit width of the domain. To further
illuminate whether the above postulate of a strong linear relation between power and variation
in the head gradient exists, we integrated power in fluid flow across the transversal extent of
the domain ($P^x_{int}$ hereafter) and plotted it against the laterally averaged head gradient (Fig. 3d-
f). In the case of unit variance, this indeed yields a strongly linear relation, with an almost
perfectly linear growth of $P^x_{int}$ with the head gradient, as indicated by the correlation coefficient
of 0.96. While this the correlation becomes weaker with increasing variance, it remains
significant with a correlation coefficient of 0.84 even for the case of $\sigma^2 = 5$. The decline in
correlation is plausible as a higher variability in $K$, in two-dimensional domains, causes stronger
transversal flow components and thus a larger deviation from the one-dimensional
heterogeneous case for which Eqs. 9 -12 are valid. As expected, the head gradients show also a
wider spread with increasing variance (Figs. 3d-f); the same holds true for power in the total
downstream fluid flow.
To check the inverse-linear relationship between $\ln(P)$ and $\ln(K)$, which was derived for the
one-dimensional approximation as well (recall Eqs. 11 - 13), we related $\ln(P^x_{int})$ to the logarithm
of laterally averaged conductivity $\ln(K_{eff})$ (Figs. 3g-i). For the unit variance case, we observe
an almost perfect linear increase of $\ln(P^x_{int})$ with a decline in $\ln(K_{eff})$, as underpinned by the
correlation coefficient of -0.92. This negative correlation declined with increasing variance to
a value of -0.81 and -0.72 for $\sigma^2 = 3$ and $\sigma^2 = 5$, respectively. Yet it is still significant, hence the
system behaves also in case of the highest variance largely similar to a heterogeneous one-
dimensional system. This is because of the confining upper and lower no-flow boundary
condition.
We thus argue that the power required to maintain the driving head difference and fluid flow in
steady state increases with increasing variance of the hydraulic conductivity field. Regions of
high power coincide with large positive deviations of the hydraulic head from its mean, and
also with "bottlenecks" of low hydraulic conductivity along the preferential pathways.



### 4.3 Entropy as a function of power and power along solute transport trajectories

Figure 5a shows the Shannon entropy at the downstream outlet $S_H(x_{max})$ as a function of the power in fluid flow integrated over the entire domain $P_{int}$ for all variance cases. The almost perfect linear decline of $S_H(x_{max})$ with $P_{int}$ reveals, in line with the gas laser example given in the introduction, that a larger power input due to a higher pumping rate leads to an higher order in the macroscale preferential transport pattern. We return to this point in section 5.3.

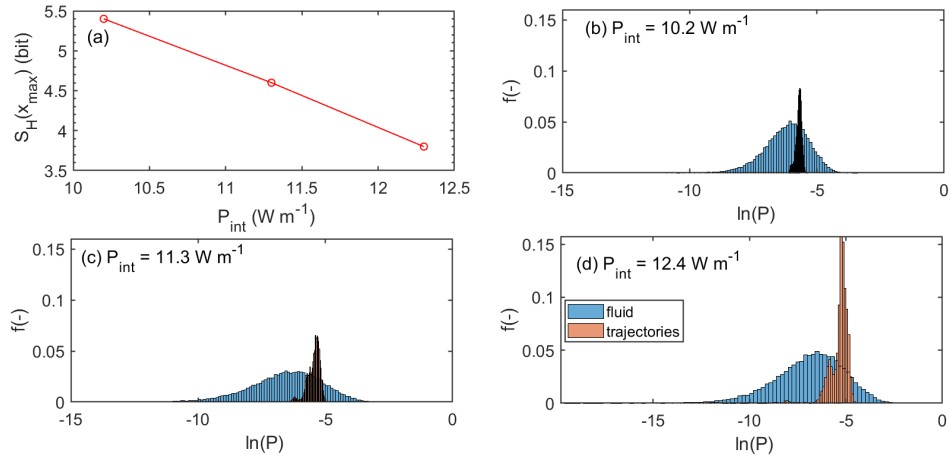

Figure 5: (a) Shannon entropy at the downstream outlet $S_H(x_{max})$ as function of the power in fluid flow integrated over the entire domain $P_{int}$ (a), cumulative distributions of $\ln(P)$ in the flow domain (blue) and of $\ln(P)$ averaged along the particle trajectories (brown) for the variance cases (b) $\sigma^2 = 1$, (c) $\sigma^2 = 3$, and (d) $\sigma^2 = 5$.

Figures 5b, c, d compare the probability density distributions (pdfs) of $\ln(P)$ within the entire flow domain (blue), against the power averaged along the actual particle trajectories (in brown, again on a log scale). While in the case of perfectly mixed flow and transport, both pdfs should be rather similar, they actually are remarkably different. The particles clearly prefer pathways of high power, as the pdfs are clearly shifted towards higher power (Fig. 5 d).

### 4.4 Space-time asymmetry and entropy export into the breakthough

To switch the observing perspective, we determined the particle breakthrough curves (BTC) for the different variances cases (Fig. 6a) and calculated their Shannon entropy as means of uncertainty and order in the arrival times, using the time step width of 0.1 dimensionless time units as bin width. The width of the breakthrough curves clearly increases with increasing

variance, indicating an earlier breakthrough, a longer tailing and a more even distribution of
normalized concentrations in time (Fig. 6 a). This implies that the Shannon entropy in arrival
times grows with increasing variance of ln($K$) reflecting a larger uncertainty and a declining
order in the temporal distribution of travel times. In this context, it is important to recall that
entropy cannot be consumed, due to the second law. This that means that the declining flow
path entropy needs to be exported from the system.

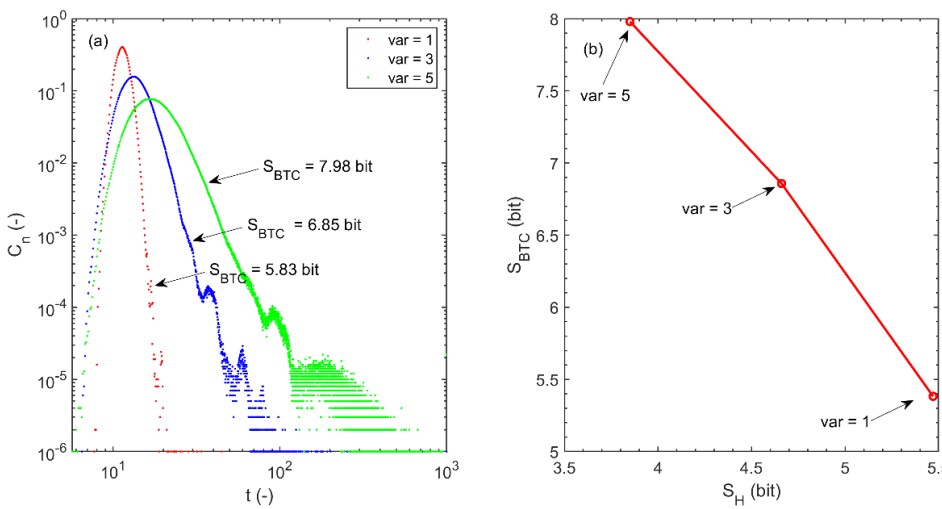


Figure 6: Breakthrough curves and their Shannon entropies $S_{BTC}$ (a); $S_{BTC}$ plotted against the
flow path entropy of the downstream outlet $S_H(x_{max})$, before particles leave the domain, for all
variance cases (b).
Figure 6b) visualizes this space-time asymmetry in entropies, the growing spatial organization
with increasing variance of ln($K$) translates due to the associated entropy export into a declining
organization in arrival times. Please note that due to the different binning in space and time,
changes in $S_{BTC}$ and $S_H$ with changing variance cannot be exactly the same. In fact, also the
entropy, which is produced due to energy dissipation, must be exported, but this is much more
difficult to quantify. The opposite of the Shannon entropy monotonies corroborate nevertheless
that reduced flow path entropy is indeed exported into the BTC. One might hence wonder
whether a perfect spatial organization due to preferential transport of the entire solute particles
through a single preferential flow path would, in the case of a step input, translate into a BTC





of maximum entropy/disorder, i.e. rectangular BTC (and vice versa). We return to this issue in
sections 5.1 and 6.

## 5 Discussion

### 5.1 An energy and entropy centered framework to characterize and explain preferential flow

This study proposes an alternative framework to quantify and explain the enigmatic emergence
of preferential flow and transport in heterogeneous saturated porous media, using concepts from
thermodynamics and information theory. We examined simulations of two-dimensional fluid
flow and solute transport based on the methods of Edery et al. (2014), and characterized the
discrete probability distribution associated with finding solute particles crossing a distinct
transversal position in a plane normal to the direction of the mean flow by means of the Shannon
entropy. In general, we found a declining entropy with increasing downstream transport
distance, reflecting a growing downstream self-organization due to the increasing concentration
of particles in preferential flow paths. Strikingly, preferential flow patterns with lower entropies
emerged when analyzing simulations in media with larger variances in hydraulic conductivity.
This implies that macro-states of higher order established, despite the higher subscale
randomness of $\ln (K)$. The key to explain this almost paradoxical behavior is the finding that
the required power to maintain the driving head difference, in steady-state flow, grows with the
variance of the hydraulic conductivity field. Due to this larger energy input, the fluid and solutes
may perform more work to increase the order in the flow path distribution, through steepening
transversal concentration gradients as reflected in lower entropies.
Notwithstanding these findings, we of course recognize that the concepts of entropy, free
energy and work are, per se, not new in hydrology. We thus place our findings in context
relative to related studies, in the sections below.

### 5.2 Measuring irreversibility and macroscale organization using the Shannon entropy

Here we show that the Shannon entropy of the transversal distribution of solutes is suited to
quantify the downstream emergence of preferential solute movement, as reflected in a declining
"flow path entropy". Lower flow path entropies and thus a stronger spatial order in preferential
transport are established when solutes are transported through stronger heterogeneous hydraulic



conductivity fields. In this context, we recall that Edery et al. (2014) analyzed breakthrough
curves using the continuous time domain random walk framework (Berkowitz et al., 2006).
When fitting an inverse power law to the breakthrough curves, the corresponding $\beta$ parameter
(which is a measure of the degree of anomalous transport, with $\beta$ increasing to 2 indicating
Fickian transport) increased with increasing variance of $\ln(K)$. Here we analyzed the Shannon
entropy of the breakthrough curves in time, and contrary to the flow path entropies, they grow
with increasing variance of $\ln(K)$. This means that higher degrees in spatial order in solute
transport that emerges at larger variances in $\ln(K)$, expressed by lower flow path entropies,
translate into a higher entropy and thus a higher disorder and thus uncertainty in arrival times.
This is reflected by an earlier first breakthrough, a retarded appearance of the peak
concentration, and a longer tailing in the breakthrough curves and higher similarity of the BTC
to a uniform, rectangular pulse. This finding coincides well with the illustrative case that
Bianchi and Pedretti (2017) used to compare solute breakthrough through ordered and
disordered alluvial aquifers.
This space-time asymmetry in entropy and organization can, however, only be explained using
the physical perspective of entropy and the second law. The emergence of spatially organized
preferential transport and the related decline in flow path entropy essentially requires an export
of the entropy from the system into the BTC. We thus conclude that the $\beta$ parameter of the
CTRW framework, is also two-fold measure for spatial organization of solute transport through
the system and temporal organization in arrival times and their asymmetry.
**5.3 Preferred flow and transport pathways as maximum power structures?**
The idea that preferential flow coincides with a larger power in fluid flow has been discussed
widely in hydrology. Howard (1971, cited in Howard, 1990) proposed that angles of river
junctions are arranged in such way that they minimize stream power; later he postulated that
the topology of river networks reflects an energetic optimum, formulated as a minimum in total
energy dissipation in the network (Howard, 1990). This work inspired Rinaldo et al. (1996) to
propose the concept of minimum energy expenditure as an enslavement principle for the self-
organized development of river networks. Hergarten et al. (2014) transferred this concept to
groundwater systems. They derived preferential flow paths that minimize the total energy
dissipation at a given recharge, under the constraint of a given total porosity and showed that
these setups allowed predictions of spring discharge at several locations. Minimum energy





expenditure in the river network implies that power therein is maximized. In this light, Kleidon
et al. (2013) showed that directed structural growth in the topology of connected river networks
can be explained through a maximization of kinetic energy transfer to transported suspended
sediments.
Our findings are in line with but step beyond these studies, which commonly refer to
preferential flow in connected, highly conductive networks. Here we find that solute particles
prefer to move through pathways of very high power, even when they are not connected by a
continuous set of cells of relatively high hydraulic conductivity. On the contrary, these
pathways incorporate regions of low hydraulic conductivity. This finding reflects the squared
dependence of power on the spatial head gradient, which in turn becomes largest in regions of
low hydraulic conductivity. We stress that this result, and our finding that a larger power input
(due to a higher pumping rate) leads to a higher order in the macroscale preferential transport
pattern, is a consequence of the imposed boundary condition. A steady-state head difference
implies a positive energetic feedback: in a real-world experiment, the pump provides this
feedback, as otherwise the gradient is depleted by the flowing fluid. Although such a positive
feedback is straightforwardly established in a numerical model by assigning the desired
constant head difference, it is important that this choice implies that such a positive feedback
exists. Due to this virtual energy input, the fluid and solutes may perform the necessary work
to rapidly pass through low conductivity bottlenecks and form an ordered preferential flow
pattern at the macroscale. The higher necessary pumping rate and energy input into the domains
with a larger variance in $K$ explain, furthermore, why preferential flow patterns of higher order
emerge with growing subscale randomness.
**6 Conclusions and outlook**
Based on the presented findings, we conclude that the combined use of free energy and entropy
holds the key to characterize and quantify the self-organized emergence of preferential flow
phenomena and to explain the underlying cause of their emergence. Information entropy is an
excellent, straightforward concept to diagnose self-organization in space and time: Here, the
formation of preferential transport is reflected in the downstream decline in the entropy of the
transversal flow path distribution and that this decline becomes stronger with increasing
variance of hydraulic conductivity. The concepts of free energy and physical entropy, however,





provide the underlying cause: steepening of transversal concentration gradients requires work,
the formation of even steeper gradients and lower flow path entropies needs even more work
and thus a higher free energy input into the open system. The higher necessary pumping rate
and energy input into the domains is the reason, why spatial organization in preferential solute
movement increased with growing subscale randomness of hydraulic conductivity. This is
behavior is very much in line with what we discussed for the gas laser in the introduction.
Entropy can, however, due to the second law not be consumed, and the declining flow path
entropy is in fact be exported from the system into the breakthrough curve. Shannon entropy
allows again for the straightforward diagnosis, while physical entropy provides the reason for
this space-time asymmetry in entropy, organization and uncertainty. Transport of all solute
particles through a single preferential flow paths implied a maximum spatial organization and
maximum/knowledge certainty about the transversal spreading of solute. However, this would,
due to the entropy export, into a maximum disorder of and thus uncertainty about the arrival
times, as the BTC would correspond to rectangular pulse of uniform concentration. Advective
diffusive transport through a homogeneous flow field implied, in case of a spatially
homogeneous step input, maximum uncertainty about transversal position of solute molecules,
while the BTC would be perfectly certain and providing minimum uncertainty about arrival
times. This space-time asymmetry in entropy implies that perfect organization and certainty
about both flow paths and travel times can never simultaneously occur. This required
consummation of entropy and thus violation of the second law of thermodynamics. However,
we wonder whether effective predictions of the entropies in the BTC and the flow path
distributions based on the knowledge driving head differences and the variance and correlation
lengths of hydraulic conductivity might be achievable in the future. This will of course not tell
us where solutes move and when they breakthrough, but predict the related uncertainty as an
important constraint of transversal distribution of transport pathways and travel times.
**Acknowledgments**
E.Z. gratefully acknowledges intellectual support by the "Catchments as Organized Systems"
(CAOS) research unit and funding of the German Research Foundation, DFG, (FOR 1598, ZE
533/11-1, ZE 533/12-1). Y.E. thanks the support of the Israel Science Foundation (grant No.
801/20);  B.B. thanks the support of the Israel Science Foundation (grant No. 1008/20) and the





Crystal Family Foundation. B.B. holds the Sam Zuckerberg Professorial Chair in Hydrology.

The authors acknowledge support by Deutsche Forschungsgemeinschaft and the Open Access

Publishing Fund of Karlsruhe Institute of Technology (KIT). The service charges for this open

access publication have been covered by a Research Centre of the Helmholtz Association.

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
