# Peer review of "Preferential Pathways for Fluid and Solutes in Heterogeneous Groundwater Systems: Self-Organization, Entropy, Work"

_Hydrology and Earth System Sciences, 2021_

## Referee Comment (RC2)

[referee-annotated manuscript omitted]

---

## Author Response (AR1)

Dear Dr. Riva,

let me first of all sincerely thank you, and both reviewers Danièle Pedrettin and Hubert Savenije, for the thoughtful assessment of our work. Attached you find the revised manuscript as a i) clean version and ii) an additional version, where all changes are highlighted in green and markups relate these changes to your and the reviewer recommendations.

The revised manuscript addresses all recommendations expect two.

- We particularly examined simulations at a 10 times lower driving head difference and thus at a 10 times lower Peclet number. Interestingly, we found an even stronger self-organization with a growing variance in K, reflected in an even stronger reduction of the flow path entropy, as for the head difference of 100 (see revised Figure 3).
- The examination of this case revealed furthermore, that the growth in power in the transversal flow component and the duration of the experiment provide an even better explanation for the growing self-organization at higher variances and at the lower head difference (see revised Figure 4 and the new Figure 5). This makes intuitive sense, a) because transversal concentrations gradients are formed by vertical flow and transport of solute particles, and b) the work performed by the fluid equals the integral power along the particle travel times. Note that the former Figure 5 is now Figure 6, and that the new Figure 5d reveals now the relation between declining flow path entropy and growing work performed by the fluid for both head differences.
- We also found the same kind of space-time anisotropy for the lower head difference: the even stronger reduction in flow path entropy with increasing variance, goes on the expense of a stronger enlarged entropy and thus uncertainty in the breakthrough curves (see revised Figure 7, Figure 6 in the previous manuscript version.)

We did not perform a separated analysis of 3D simulations, because we do not expect qualitatively new insights. As explained in line 672 - 678, we expect similar behavior, because the local changes in power of the transversal flow component arise from the local feedback on the pressure head gradient upstream of the low conductivity bottlenecks. The gradients steepen ahead of these bottlenecks, which implies a higher power in the transversal flow component. This feedback will also occur in a 3D confined system, as it is a direct result of the boundary conditions (no flow boundary conditions for the upper, lower, inlet and outlet boundaries).

We also streamlined the manuscript, removed duplications and changed equations, as requested. While we truly appreciate your recommendation to restructure the manuscript, we kept sequence to introduce first the numerical simulations in section 2 and then the concepts Entropy, Power and Work in section 3. We think this improves the clearness of the presentation, because these simulations form the main motivation of this study for the theory in section 3. This is mentioned in line 181 and further explained in lines 222-244 (and highlighted by the revised Figure 1).

I hope you forgive that I will did not copy our detailed replies to both reviewers in the annex, as these are provided in the discussion.

Best regards,

Erwin Zehe